# Intelligent Wireless Charging Path Optimization for Critical Nodes in Internet of Things-Integrated Renewable Sensor Networks

**DOI:** 10.3390/s24227294

**Published:** 2024-11-15

**Authors:** Nelofar Aslam, Hongyu Wang, Muhammad Farhan Aslam, Muhammad Aamir, Muhammad Usman Hadi

**Affiliations:** 1School of Information and Communication Engineering, Dalian University of Technology, Dalian 116024, China; whyu@dlut.edu.cn; 2Department of Economics, The National College of Business Administration and Economics, Multan 61000, Pakistan; farhan_aslam38@yahoo.com; 3School of Sciences, Harbin Institute of Technology, Shenzhen 518060, China; haamir99@stu.hit.edu.cn; 4School of Engineering, Ulster University, Belfast BT15 1AP, UK; m.hadi@ulster.ac.uk

**Keywords:** renewable wireless sensor networks, ant colony optimization, wireless power transfer, Internet of Things, wireless sensor networks

## Abstract

Wireless sensor networks (WSNs) play a crucial role in the Internet of Things (IoT) for ubiquitous data acquisition and tracking. However, the limited battery life of sensor nodes poses significant challenges to the long-term scalability and sustainability of these networks. Wireless power transfer technology offers a promising solution by enabling the recharging of energy-depleted nodes through a wireless portable charging device (WPCD). While this approach can extend node lifespan, it also introduces the challenge of bottleneck nodes—nodes whose remaining energy falls below a critical value of the threshold. The paper addresses this issue by formulating an optimization problem that aims to identify the optimal traveling path for the WPCD based on ant colony optimization (WPCD-ACO), with a focus on minimizing energy consumption and enhancing network stability. To achieve it, we propose an objective function by incorporating a time-varying *z* phase that is managed through linear programming to efficiently address the bottleneck nodes. Additionally, a gateway node continually updates the remaining energy levels of all nodes and relays this information to the IoT cloud. Our findings indicate that the outage-optimal distance achieved by WPCD-ACO is 6092 m, compared to 7225 m for the shortest path and 6142 m for Dijkstra’s algorithm. Furthermore, the WPCD-ACO minimizes energy consumption to 1.543 KJ, significantly outperforming other methods: single-hop at 4.8643 KJ, GR-Protocol at 3.165 KJ, grid clustering at 2.4839 KJ, and C-SARSA at 2.5869 KJ, respectively. Monte Carlo simulations validate that WPCD-ACO is outshining the existing methods in terms of the network lifetime, stability, survival rate of sensor nodes, and energy consumption.

## 1. Introduction

The broad integration of the Internet of Things (IoT) and wireless sensor networks (WSNs) is getting more attention these days for data monitoring and tracking [1,2]. Smart sensor nodes with computing, sensing, and storage capabilities make up the WSN. Due to their small battery capacity, these sensor nodes only have a limited amount of time before they quit working. To solve this problem, ambient energy harvesting has become the focus of much research [3]. We can convert ambient energy from wind, air, and solar into electrical energy to recharge the batteries of sensor nodes [4]. However, the elementary limitations of these approaches are time-variable and environment-dependent, which makes them unreliable [5]. WSNs have shown remarkable lifetime expansion over the previous few years because of wireless charging to eliminate the dependency on wires and manual battery replacement [6].

One outstanding method for wireless energy replenishment that finally leads to the WSNs’ lifetime enrichment is wireless power transfer (WPT) [7]. The topical research focuses on recharging the sensor nodes by deploying the wireless charging vehicles. These wireless charging vehicles are equipped with WPT devices and large-capacity batteries. Most of the charging schemes are on-demand, which can transfer energy through a static source to make the WSN perpetual [8]. The latest research work emphasizes deploying the wireless charging drone to charge the sensor nodes in a rechargeable wireless sensor network. However, the charging drones have a limited amount of energy to transfer the sensor nodes due to their size as compared to the charging vehicles.

We can categorize the previous research on wireless charging the sensor nodes into five distinct groups:A single drone serves as a wireless charger [9];Predefined trajectory and schedule for wireless chargers [10];Multiple chargers (e.g., drones, wireless charging vehicles, and wireless charging vehicle-carried drones) for a network [11];Multi-objective charger (e.g., charger and base station) [12,13];Power balancing and collaboration of manifold wireless chargers [14,15].

Drawing from prior research, all approaches aim to improve the lifespan and reduce the energy consumption of mobile chargers, albeit with certain limitations. Distinctively, our focus is on the efficient routing path of the wireless portable charging device (WPCD) to recharge the node’s battery with the least remaining energy first. Additionally, we aim to improve awareness of the remaining energy of each node by enabling the IoT connection. This factor helps to prolong the lifetime of a large-scale network because there is no threshold assumption about the remaining energy of the nodes. We previously assumed that the threshold value represents the minimum energy required to maintain the node’s operation, but this assumption fails when dealing with a bottleneck node [16,17]. Bottleneck nodes are those with extremely low remaining energy (e.g., even lower than the threshold value). The IoT cloud connects to the renewable wireless sensor network (RWSN) to share each sensor node’s energy status. The constant exchange of energy status at every node makes the WPCD proficient.

We can summarize this work’s key contribution as follows:To ensure the perpetual life of the RWSN, we leverage a WPCD to recharge the sensor node as well as the whole network by using wireless power transfer technology (WPT). We formulate a numerical objective function by segmenting the nodes into different phases. The introductory *z* phase indicates the lower remaining energy nodes as bottleneck nodes; thus, a WPCD recharges those nodes on a priority basis.We also propose a novel route for the WPCD to traverse all the nodes including bottleneck nodes during the entire charge cycle. The proposed optimal routing trail is defined as WPCD-ACO and handles the optimization of travel cost in terms of distance and travel time with the help of a modified ant colony optimization (ACO) algorithm.We also design a connection of the IoT with the RWSN through a gateway sensor node, which is constantly exchanging the remaining energy information of every node. This energy information makes the WPCD more competent to recharge the sensor nodes promptly.Finally, this joint optimal solution of the objective function and WPCD-ACO path prolongs the lifetime, ensuring stability, and reduces the energy consumption of the network even at large-scale deployment of the nodes.

The research paper arranges its remaining content as follows: Section 2 delineates the relevant aspects of the proposed research. Section 3 defines the proposed system model and formalizes the RWSN’s optimal charging schedule. Section 4 presents the simulation results and graphs. Lastly, Section 5 concludes the whole research work.

## 2. Related Work

This section narrates a comprehensive literature review of the WPT technology for charging the sensor nodes, WPCD routing, and the IoT-enabled WSN to understand the state-of-the-art research.

### 2.1. Wireless Power Transfer (WPT)

Most of the existing work uses wireless power transfer technology to recharge the sensor nodes via single or multiple chargers. Zhong et al. use this breakthrough innovation to charge and collect data in the WRSN. They proposed an optimized bifunctional vehicle trail for charging [18]. The wireless energy transfer (WET) technology was also explored by Tomar et al. for recharging the sensor nodes. They proposed a fuzzy logic charging approach of a mobile charger (MC) based on the consideration of residual energy, distance, and node density. They further expanded their research in terms of numerous MCs in a network [19]. Similarly, in [20], a joint routing and charging algorithm (J-RCA) was proposed by Lu et al. They devised multiple MCs for data collection by considering the topology changes of wireless rechargeable sensor networks to prolong their lifetime. Jia et al. presented a joint energy harvesting and decreasing energy consumption in the wireless rechargeable sensor network. Their algorithm focuses on the trade-off between two factors. One is reducing the distance of the MC and the other is the charging efficiency among the nodes [21].

While all existing methods use WPT for extending the lifetime of the WSN, they overlook the survival rate of sensor nodes. Subsequently, the risk of bottleneck node failure may increase.

### 2.2. Wireless Charger’s Routing

Another line of research has focused on maintaining the wireless charger’s residual energy to distribute the charging resources equally within the entire network. The team of Aslam et al. has proposed the shortest path algorithm to reduce the path of the WPCD, thus enhancing energy conservation among sensor nodes [16]. They extended their work by proposing the online clustering-state action reward state action (C-SARSA) mechanism for fair energy consumption between the RWSN [17]. On the contrary, Huang et al. proposed an energy consumption balanced tree construction minimizing the event missing rate (ECBT-MEMR) method to foresee the energy consumption of each sensor node. Afterward, this energy consumption value is used to plan the charging tasks [22]. Additionally, Liang et al. proposed an improved firefly algorithm (IFA) for increasing the charging proficiency and saving the energy consumption of the network. They optimize the deployment of MCs utilizing IFA [23]. Tao et al. find the problem of charging efficiency and coverage in the RWSN. They maximize the optimization problem to sort the charging layout and scheduling strategy [24].

However, the state-of-the-art approaches only focus on the performance of the wireless charger’s routing without considering the data transmission rate of each node. As a consequence, all the algorithms may not properly work in the case of high data transmission, which eventually leads to the dynamic energy consumption of the node (bottleneck node).

### 2.3. IoT Enabled WSN

The majority of the research on IoT-enabled WSNs is concentrated on particular tasks such as smart city development [25], healthcare monitoring [26], greenhouse [27], military surveillance [28], etc. Metia et al. formulated the monitoring of air pollution with the help of the IoT. They proposed a model of extended fractional-order Kalman filtering (EFKF) with IoT integration to collect more reliable and accurate data from the air [29]. A team of Rahman et al. also implemented the IoT-equipped WSN to monitor the environment. They implemented the low-power, low-cost, and wide-range (LPWR) framework using the low-power and low-range (LoRa) WSN [30].

In [31], an event-driven IoT-enabled WSN is formulated to handle privacy preservation. They evaluate the simulation results by a chessboard alteration pattern (SLP-ED-CBA), source location privacy for event detection (SLP-ED), and grid-based source location privacy (GB-SLP). In [32], the authors designed an IoT architecture for ensuring fault-tolerance data routing within the wireless sensor networks. They deployed the Q-learning technique to route the data from the cluster head to the mobile sink node. This reinforcement learning technique guarantees the maximization of lifetime and minimization of data loss and energy consumption.

To the best of my knowledge, all the previous research is presented particularly for extending the network lifetime or IoT enabling the wireless sensor network. This paper aims to settle the perpetual life of the RWSN by connecting it to the IoT. The incorporation of the IoT with RWSN makes the WPCD more accurate while defining the charging pattern of sensor nodes even in the case of bottleneck nodes. This charging outline is formulated by utilizing the residual energy and distance among each sensor node. The proposed structure is composed of a numerical objective function and the WPCD-ACO algorithm.

## 3. System Roadmap

To boost the longevity and large-scale deployment possibilities of WSNs, energy harvesting is an auspicious solution. The proposed architecture considers a 1000 m × 1000 m square area in a two-dimensional space that is more flexible and less expensive. We randomly deployed N numbers of static nodes in the designated area. A WPCD is provisioning power to the node’s batteries through WPT technology. The energy level of apiece node *i*’s battery must fall under the range of $E_{max}<={node\;i’s\;battery>=E}_{min}$ to keep them operational. Initially, all the nodes are fully charged with maximum energy, $E_{max}$. There is a fixed base station (BS) and a rest station (RS) in the field. The WPCD initiates its expedition from the RS to replenish every node’s battery until it recharges its energy level to $E_{max}$. The threshold $E_{min}$ refers to the minimum energy required by a node to continue its functions. Each node contains a nickel-metal hydride (NiMH) battery, outfitted with a power-receiving coil. The nominal voltage of 1.2 V and a quantity of electricity of 2.5 Ah from the battery aid in the calculation of the values of $E_{max}$ and $E_{min}$ as shown in Equations (1) and (2):(1)$$E_{max}=3600\;s\times 1.2\;V\times 2.5\;Ah=10,800\;J=10.8\;KJ$$
(2)$$E_{min}=0.05\times E_{max}=540\;J=0.54\;KJ$$

A gateway node connects the entire wireless sensor network to the internet cloud. This gateway node exchanges the remaining energy information of each node with the IoT cloud. The WPCD accesses the remaining energy values from the cloud before starting its journey. This IoT connection in the WPCD adjusts the power transfer for each node, which enhances the energy efficiency of the charging procedure. The round trip of the WPCD in a single charge cycle of the RWSN is called the Hamiltonian cycle. This Hamiltonian cycle represents that the WPCD needs to recharge its battery through a static source of energy at the RS. During instantaneous circumstances, the WPCD can be a driver-operated vehicle. This type of vehicle is more intellectual, as a driver can prevent the WPCD from facing hurdles on its journey. There is pioneer WiTricity Corporation which manufactures lightweight and low-power wireless charging products. For example, WiTricity-3300 is easy to install on the WPCD, having a size of 20 cm × 28 cm × 7 cm and a weight of 3600 g [33]. For this research, the fuel and maintenance costs of the WPCD are considered out of scope.

Figure 1 represents the understanding of the whole scenario. Simultaneously, Table 1 summarizes the notations used in the proposed system model.

At the moment, sensor nodes start generating data at a rate of $R_{i}$, and their energy consumption level gradually increases. This perpetual decrement in the node’s battery continues till the threshold value, $E_{min}$. On the contrary, the nodes that are experiencing high data transmission and undergoing a fast depletion of energy occasionally face the lower remaining energy level $E_{xmin}$, below the threshold value. All these types of sensor nodes with a lower remaining energy level are called bottleneck nodes, *Nb*. The WPCD must provide the charging priority to the bottleneck nodes by switching at the *z* phase. Therefore, the WPCD starts its charging cycle from node *i*, or the bottleneck node according to the proposed WPCD-ACO. The traversing velocity of the WPCD is V (m/s). The approaching time of the WPCD at node *i* is denoted as $a_{i}$ to recharge its battery at the power transmission rate of U. The time spent by the WPCD to recharge the battery of any node is abbreviated as *τ_i_*. After recharging the battery of the first node, the WPCD traverses towards the next node *j*, and so on. At last, the WPCD comes back towards the RS to recharge its own battery. Thus, the time spent by the WPCD in the RWSN is called field time, *τ* while the time spent at the RS is defined as vacation time, *τ_vac_*.

### 3.1. Formulation of Numeric Objective Function

At the onset, we assumed that the battery of nodes would never fall below $E_{min}$. To be immune from this assumption, we have formulated our optimization problem through linear programming. Let us suppose σ∗ is the solution of our optimization function. The solution can be achieved by considering the *z* phase in the charge cycle of the WPCD. During phase *z*, the focus is on any bottleneck node that has a remaining energy lower than $E_{min}$.

**Theorem** **1.**
*An optimal solution with a phase z recharges the bottleneck node first during the realization of the charge cycle.*


**Lemma** **1.***Suppose a dynamic problem* σ∗ *is constructing z phase with two different sets upon its remaining energy level*, i.e., *prior (*$<E_{min}$*) and allowed (*$\geq E_{min}$*).*

**Proof** **of** **Theorem 1.**The RWSN incorporates the WPCD to power its nodes. The nodes with high data transmission can lose their energy faster. Therefore, despite considering the different numbers of WPCDs, it is easy to consider different phases. Firstly, we calculate each node’s remaining energy status to classify them into either the normal phase or the *z* phase. Next, the gateway node uploads the remaining energy information from the nodes to the IoT cloud. The WPCD can directly access the remaining energy information of the whole RWSN before starting the charging process. The normal phase is in charge for charging the normal energy nodes whereas the *z* phase is responsible for charging the lower energy nodes. There are two different sets, known as the prior and allowed sets, that maintain the remaining energy levels. The prior set holds the nodes whose remaining energy level is less than the threshold value $E_{min}$. In contrast, the allowed set includes the nodes with a remaining energy level greater than $E_{min}$. □

### 3.2. Data Transmission and Energy Consumption in WPCD-ACO Scenario

The objective function σ∗ is formulated at a time $\tau_{z}$ for handling the *z* phase. Furthermore, the σ∗ function also handles the data transmission and energy consumption in the proposed WPCD-ACO scenario, which is computed as follows:



$Maximize:\sigma *=\frac{\tau_{vac}}{\tau }$


*Subject to:*




(3)
$$\sum_{k\in N}^{k\neq i}g_{ki}\left[z\right]+R_{i}=\sum_{j\in N}^{j\neq i}g_{ij}\left[z\right]+g_{iB}\left[z\right]$$



Herein, the rate of data transmission from node *i* to *j* and *i* to the BS is denoted as $g_{ij}\left[z\right]$ and $g_{iB}\left[z\right]$ during phase *z*.$\;R_{i}$ and $g_{ki}\left[z\right]$ are the rate of data generated by node *i* and the rate of data gathering from any other node *k*. The rate of energy consumption $p_{i}$ during data transmission is handled by (4).
(4)$$p_{i}\left[z\right]=\sum_{k\in N}^{k\neq i}\rho \times g_{ki}\left[z\right]+\sum_{j\in N}^{j\neq i}C_{ij}{\times g}_{ij}\left[z\right]+C_{iB}\times g_{iB}$$

ρ is the energy consumption coefficient and $g_{ki}\left[z\right]$ is the energy consumption during data receiving from node *k*. The rate of energy consumption for data sending along with its coefficients is denoted as $C_{ij}{\times g}_{ij}\left[z\right]+C_{iB}\times g_{iB}\left[z\right]$ during the *z* phase. In the case of the bottleneck node, when the WPCD arrives at node *i*, its remaining energy can be lower than $E_{min}$ as in (5).
(5)$$E_{xmin}\left(a_{i}\right)\leq E_{min}$$

$E_{xmin}$ is the remaining energy of the bottleneck node, and (6) is responsible for the measure $E_{xmin}$ of node *i* during the *z* phase. The *z* phase is divided into small intervals of time *z* as formulated in Equations (6)–(8).
(6)$${\;\;\;E}_{xmin}\left[z\right]=E_{xmax}-{C_{ij}\times g}_{ij}\left[z\right]+C_{iB}\times g_{iB}\left[z\right]$$
(7)$$\sum_{z\in Z}^{z\neq i}\tau_{z}p_{i}\left[z\right]\leq E_{max}-E_{min}$$
(8)$$\tau =\sum_{z=1}^{Z}\tau_{z}$$
where $\tau_{z}$ is the set of whole-time instances during the *z* phase, and τ is the overall time spent by the WPCD in a charge cycle.
(9)$$D_{ij}=\sqrt{{\left(j_{2}-j_{1}\right)}^{2}+{\left(i_{2}-i_{1}\right)}^{2}}$$
(10)$$\sum_{i=1}^{N}D_{c}=\sqrt{\sum_{i=1}^{N}{\left(N_{2}-N_{1}\right)}^{2}+{\left(i_{2}-i_{1}\right)}^{2}}$$

Equations (9), and (10) are computing the distances between the node *i* to *j* and the entire nodes as well. The arrival time of the WPCD at each node is formulated in (11).
(11)$$a_{i}=\tau +\sum_{i=1}^{N}\frac{D_{c}}{V}$$limit to
$$g_{ij}\left[z\right]\geq 0,\;{\;g}_{iB}\left[z\right]\geq 0,\;p_{i}\left[z\right]\geq 0\;\left(i,j\in N,\;i\neq j,\;z\in Z\right)\;\tau_{z}\geq 0\;\left(z\in Z\right)\;\tau ,\;\tau_{vac}\geq 0\;\left(i\in N\right)$$

### 3.3. Uploading and Accessing the Energy Level Information to IoT from RWSN

In an IoT-enabled WSN, the storing of battery-level data requires cloud computing, which can facilitate long-lasting storage. Figure 2 visualizes the suggested image of the proposed scheme:

Let us suppose the data generation rate of sensor node *i* is $G_{i}\left(t\right)$ at a time t. Then the total data generated by sensor nodes during the time interval [0, T] is measured as follows in (12):(12)$$G_{i}\left(T\right)=\sum_{i=1}^{N,\;\;\;i\neq j}\int_{0}^{T}G_{i}\left(t\right)dt,\;\;\;\;\;\;\;\;\;\;\;\;\;\;\;\;N_{b}\in N$$

While data transmission and energy consumption demonstrate the energy level of every sensor node inside the network from (3) to (8). Battery level information for sensor node *i* at time t is denoted as ${BL}_{i}\left(t\right)$. The gateway node assists in uploading the battery level information from the sensor node *i* to the IoT cloud at time t at the rate of ${µ}_{i}\left(t\right)$. The comprehensive model computes from (13) to (20) over a time interval [0, T] as shown below:(13)$${BL}_{i}\left(T\right)=\int_{0}^{T}{BL}_{i}\left(t\right)\;dt$$
(14)$${µ}_{i}\left(t\right)={BL}_{i}\left(t\right)$$
(15)$$X_{i}\left(t+1\right)=X_{i}\left(t\right)+\sum_{i=1}^{N,\;\;\;i\neq j}{BL}_{i}\left(t\right),\;\;\;\;\;\;\;\;\;\;\;\;\;\;N_{b}\in N$$
(16)$$X_{i}\left(T\right)=X_{i}\left(0\right)+\int_{0}^{T}\sum_{i=1}^{N,\;\;\;i\neq j}{BL}_{i}\left(t\right)dt,\;\;\;\;\;\;\;\;\;\;\;\;\;N_{b}\in N$$
(17)$$M\left(t\right)=\sum_{WPCD=1}^{N,\;\;{\;N}_{b}\in N\;}M_{WPCD}\left(t\right)$$
(18)$$M\left(T\right)=\int_{0}^{T}\sum_{WPCD=1}^{N,\;\;N_{b}\in N}M_{WPCD}\left(t\right)dt$$
(19)$$r_{WPCD}\left(t\right)=f\left({BL}_{i}\left(t\right)\right),\;\;\;\;\;\;\;\;\;\;\;\;\;\;\;\;\;\;\;\;\;\;\;\;\forall_{i,\;\;\;WPCD}$$
(20)$$r_{WPCD}\left(T\right)=\int_{0}^{T}f\left({BL}_{i}\left(t\right)\right),\;\;\;\;\;\;\;\;\;\;\;\;\;\;\;\;\;\forall_{i,\;\;\;WPCD}$$

$X_{i}\left(t+1\right)$ and $X_{i}\left(T\right)$ are the change in cloud storage (battery information of sensor node *i*) and cloud storage (battery information of sensor node *i*) at a small interval of time t and total time T, respectively. Likewise, $M\left(t\right)$ and $M\left(T\right)$ are the battery information access rate for the WPCD at different times. To finish, $r_{WPCD}\left(t\right)$ and $r_{WPCD}\left(T\right)$ are the retrieval rate of battery information by the WPCD from the IoT cloud at time t and T, correspondingly. The WPCD can easily access and retrieve the battery level information of sensor nodes from the IoT cloud and make a decision to visit the node in the simple or *z* phase of the RWSN.

### 3.4. Ant Colony Optimization

Ant colony optimization (ACO) is a probabilistic and meta-heuristic method to find the solution to combinatorial problems [34]. The nature of the proposed problem is also combinatorial and relies on ACO. ACO has great strength in finding the optimal path at a large-scale area, which is similar to our formulated problem [35]. Therefore, a large number of artificial ants are created as computational agents for the problem at hand. At each iterative stage, an ant goes from state *i* to state *j* referred to as intermediate solutions. Initially, there are no pheromone trails; thus, half of the ants will move for one solution, and another half will explore an additional one randomly. An ant *k* computes the solution between states *i* and *j* and maintains a set of feasible expansions, $A_{k}\left(i\right)$. Subsequently, Equation (21) calculates the probability of the ants to move among states:(21)pijk=тijα×ƞijβ∑q∈allowedk×тiqα×ƞiqβ,          j∈allowedk                                                                                        0                                                                        otherwise
where ${т}_{ij}$, α≥0 are the pheromone trails from state *i* to *j*, and parameters to regulate the influences of ${т}_{ij}$, respectively. ${ƞ}_{ij}$ is the desirability of changing the states defined as $\frac{1}{d_{ij}}$, and β≥1 is a parameter to regulate the effect of ${ƞ}_{ij}$, whereas ${ƞ}_{iq}$ and ${т}_{iq}$ show the effectiveness and level of pheromone trails for the transition to other possible states. ${allowed}_{k}$ is the set of states that needs to be explored next. Once the solution is complete, the trail levels are updated as in (22) and (23).
(22)$${т}_{ij}\leftarrow \left(1-\rho \right)\times {т}_{ij}+\sum ∆{т}_{ij}^{k}$$
(23)$$∆{т}_{ij}^{k}=\left\{\begin{array}{c} \frac{Q}{L_{k}},\;\;\;if\;ant\;k\;moves\;on\;the\;edges\;ij\\ 0\;\;\;\;\;\;\;\;\;\;\;\;\;\;\;\;\;\;\;\;\;\;\;\;\;\;\;\;\;\;\;\;\;\;\;\;\;\;otherwise \end{array}\right.$$
where ρ∈0, 1 handles the pheromone evaporation and $∆{т}_{ij}^{k}$ is the amount of pheromone deposited by *k* ant. Q and $L_{k}$ are the constant and length (cost) of the ant’s journey.

### 3.5. Proposed Optimum WPCD-ACO Path

In the proposed algorithm, states are defined as N number of nodes that are going to be traversed by the WPCD. The numerical formulation of the proposed optimum path for the WPCD is called WPCD-ACO. At time = 0, these nodes are classified into two phases based on the following specific rules:

Every node is served only once in a single charge cycle of WPCD;The high remaining energy nodes have a low possibility of being served by the WPCD in every charge cycle;After the completion of one charge cycle, if the tour is short, then ants lay out more pheromones on the entire traversed charging route;The path that has the most intense pheromone will have a probability of being selected;After every charge cycle, the pheromone trails evaporate.

These phases contain different sets of nodes, such as ${prior}_{k}\left(need\;to\;recharge\;first\right)$ and ${allowed}_{k}$ (can be recharged afterward).

**Theorem** **2.**
*In the proposed intuition, if two nodes are neighbors, they have a short distance, and a high data transmission rate must belong to the same phase z.*


**Proof** **of** **Theorem 2.**The ratio of low remaining energy and less distance are the factors that should be taken into account for the next destination of an ant. Thus, the nodes with the minimum ratio fall into the same set of ${prior}_{k}$ from phase *z*. For instance, a node *j* is grouped into the phase *z*, denoted as *z (j)*. In that case, the probability of node *j* being visited after node *i* by the ant *k* is calculated from (21), rewritten by (24) to (26).

(24)
pijk=  тijα×ƞijβ×rikzj∑q∈allowedk×тiqα×ƞiqβ×[rikzq],        j∈allowedk                                                                                             0                                                                                       otherwise


(25)
$$r_{i}^{k}z\left(j\right)=\frac{E_{xmin}}{D_{ij}}$$


(26)
ritk=∑s∈ztтisα×ƞisβ∑q∈allowedk×тiqα×ƞiqβ,     zt∈priork                                                                                      0                                                                otherwise

The $E_{xmin}$, $D_{ij}$ can be calculated through the aforementioned Equations (6) and (9). Obviously, $r_{it}^{k}z$ is the minimum ratio of remaining energy and distance from the *z* phase. On the other side, the updating rule for pheromone trails is also formulated again as (27) to (30).
(27)$${т}_{i\;N}\left(t\right)=\sum_{i=0}^{N}D_{c}+c$$
(28)$${т}_{ij}\left(t+1\right)=\rho \times {т}_{ij}\left(t\right)+∆{т}_{ij}$$
(29)$$∆{т}_{ij}=\sum_{k=1}^{A}∆{{т}_{ij}}^{k}$$
(30)∆тijkt+1=QLk,    if ant k travels from node i to j                                                                    0                                        otherwise
where *t* and *c* are the counter of iteration and mean of distances, respectively. *t +* 1 is the next iterative level and *h* is the moving speed of ant *k* in the field. □ 

To understand the working of the proposed optimum WPCD-ACO path, Algorithm 1 is delineated below:
**Algorithm 1.** Optimum WPCD-ACO Path for WPCD**Sr. No.**

**Input:** The number of ants *k*, α is the parameter to regulate the influences of ${т}_{ij}$, ${т}_{ij}$ are the pheromone trails from node *i* to *j*, β is a parameter to regulate the effect of ${ƞ}_{ij}$, ${ƞ}_{ij}$ is the desirability of changing the nodes, ρ is the evaporation rate of pheromone trails and *h* is the speed of ants *k*.
**Output:** WPCD Optimal Path, **D_c_**1.Empty the value of **D_c_**.2.**Initialize:** Total number of ank *k*, *A* = 20 or 30, *α* = 1, β=1, ρ=0.7     Q = 1, *h* = 5 cm/s, time = 0, *c* = 0.0014, *T_ij_* = *c*, and Δ*T_ij_* = 0     time is the counter of Time. For every node *T_ij_* = *c* is the path density.3.Initialize all the nodes and their locations.    **Set**, s = 0: 4.**for** *k* == 1 && *k* ∈ *A* **do**    // s is the counter of the ant’s step. Ant *k* chooses the random nodes and add the nodes to the set of **visited*_k_***. Then add this in the group of **prior*_k_*** nodes.5.**Repeat** till s ≤ m     // m is the path from one node to another.  s = s + 1 6.    **for**
*k* = 1 && *k* ∈ *A* **do**        Select the next node to visit with the help of probability from Equations (24) to (26).         Ant *k* will travel to the selected node. Add the selected node into the **visited*_k_***        which eventually adds into the set of **allowed*_k_*** or **prior*_k_***.7.        **for**
*k* = 1 && *k* ∈ *A* **do**            Move the ant *k* between the **allowed*_k_*** or **prior*_k_*** to traverse all nodes for their battery replenishment.8.Calculate the length of the journey ***L**k*** traveled by the ant *k*.Update the optimal Path.9.           **for** *k* = 1 && *k* ∈ *A* **do**               Update the **T*_ij_*** pheromone trail by Equations (27)–(30).           **end for**       **end for**    **end for**10.time = time + 111.**If** time ≤ Time **   then**       Nullify all the set of **allowed*_k_***, **visited*_k_*** or **prior*_k_***12.Go to step 2.13.**else**14.Compute the optimal path, D_c_15.**end for**

## 4. Results

To divulge a better understanding of the proposed scheme, we simulated a series of results with different numbers of iterations and ants. The proposed WPCD-ACO is compared with the state-of-the-art literature to demonstrate its effectiveness even in the case of the bottleneck node. 

Table 2 lists the parameters used in the simulation application scenario to study the influence of ants and pheromone trails.

Simulation starts with the communication of data among the sensor nodes. Figure 3 shows the data-sending rate of the bottleneck and other nodes as well.

Data communication within the network consumes a significant amount of energy. Hence, an increase in data transmission leads to an increase in energy consumption, which in turn reduces the remaining energy even below the threshold value, $E_{min}$. As soon as the remaining energy of any bottleneck node reaches $E_{xmin}$, this node switches to the *z* phase and recharges its battery via the WPCD on a priority basis to remain operational forever. The WPCD transmits power to the nodes of the *z* phase first. We validated the results with the help of a Monte Carlo simulation, where c is the mean distance matrices of the nodes. The proposed optimum path for WPCD-ACO is implemented through the influential behavior of ant colony optimization satisfying the *z* phase effect for drifting the WPCD. Hence, the reward function of the WPCD-ACO algorithm is tested at different values of ρ. Figure 4 shows that ρ=0.7 is formulating the best reward for the ant’s behavior in the WPCD-ACO algorithm.

The novel objective function $\sigma *=\frac{\tau_{vac}}{\tau }$ is showing the successful application of the *z* phase and maximizes the ratio of vacation time to field time. Moreover, if the charging methodology effectively extends the node’s lifetime, it should result in a reduction in the WPCD influx time. Figure 5 gives evidence that the arrival time of the WPCD at each node is significantly lower than the benchmark algorithm in the literature. The lower arrival time undoubtedly indicates that the IoT-enabled WSN is providing more intellectual information about the remaining energy to the WPCD. Based on the provided energy, the proposed WPCD-ACO measures the optimum path for recharging the nodes, which reduces its arrival time instantly.

The proposed method calculates the total travel time by comparing it with the shortest path, grid clustering, and Dijkstra algorithm. Figure 6 clearly states that the proposed methodology has less travel time than the other equivalent literature. The lower traveling time in the field enables the WPCD to spend more time at the rest station for its vacation.

Likewise, one of the core objectives of the proposed scenario is to decrease the travel distance of the WPCD in the field. Figure 7 illustrates that the outage optimal distance measured by the proposed WPCD-ACO method is 6092 m as compared to 7225 m in the case of the shortest path, 6761 m in the case of grid clustering, and 6142 m in the case of Dijikstra’s algorithm, respectively [19,21].

The grid clustering tactic, as presented in [19], comprises a wireless charging device to supplement the power of sensor nodes. This approach effectively increases the network lifetime and significantly reduces its energy consumption. Other research in [19,22] also introduces the wireless power transmission approach to save the network’s lifetime. The comparative scrutiny of Figure 8 displays that our methodology conserves energy more efficiently than the others. The energy consumption is almost 50 percent lower than previous approaches. Quantitatively, the energy consumption for a single hop is 4.8643 KJ while GR-Protocol has 3.165 KJ, Grid clustering has 2.4839 KJ, C-SARSA has 2.5869 KJ, and the proposed WPCD-ACO has 1.543 KJ, correspondingly.

Finally, to validate the superiority of our proposed WPCD-ACO, we have compared our work with that of [33]. In line with the fundamental objective of the WPCD-ACO methodology, bottleneck nodes are being recharged on a priority basis. Therefore, the survival rate of the nodes is almost 99 percent in the entire network. In contrast, ECBT-MEMR has a lower number of surviving nodes as compared to WPCD-ACO at a different number of deployed nodes as shown in Figure 9.

## 5. Conclusions and Future Directions

Harnessing the abilities of WPT technology has arisen as a means of combating battery lifespan limitations and enhancing the functional lifetime of the IoT-integrated WSNs. In this study, we investigated the numerical optimization of a charging trail of a WPCD, especially in the case of a bottleneck node whose remaining energy is at an alarming stage. WPT restores the energy of sensor nodes, enabling us to achieve the optimal distance and lower traveling time of the WPCD. Once the optimization of the objective functions has been planned, an iterative method WPCD-ACO, based on the ACO, is chosen along with the IoT.

The Monte Carlo simulation is performed to corroborate the significant numerical results as follows:

The long-lasting sensor network under the WPCD-ACO algorithm is more efficient than the compared shortest path, grid clustering, and Dijikstra’s algorithm. It is achieved by calculating the optimal distance of the WPCD, reduction in arrival time of the WPCD, and lower travel time of the WPCD within the RWSN.

The rate of surviving nodes in the proposed WPCD-ACO method is adequately higher than the other compared methods like single hop, grid clustering, and ECBT-MEMR. It is due to handling the bottleneck nodes in the *z* phase of the optimization.

The energy consumption of the WPCD-ACO algorithm is less than the single hop, GR-Protocol, grid clustering, and C-SARSA algorithms due to a better awareness of the energy level information of each sensor node with the help of an IoT connection.

Furthermore, the ongoing research on wireless power transfer techniques should be more versatile for energy transfer, as well as support expanding the coverage area of wireless charging structures.

## Figures and Tables

**Figure 1 sensors-24-07294-f001:**
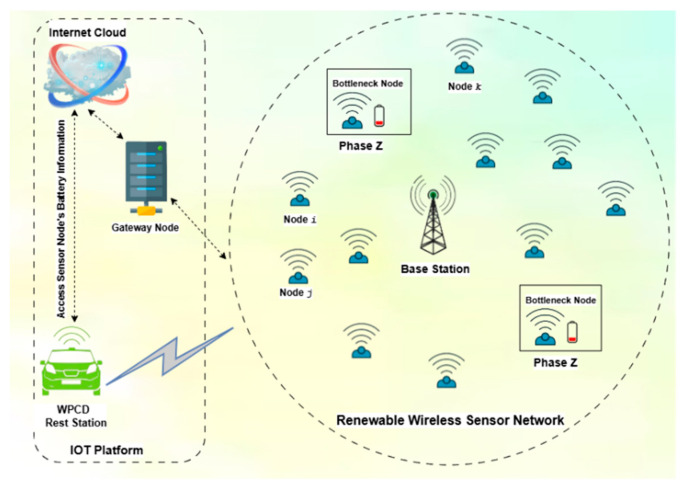
The layout of the IoT-RWSN with an effect of phase *z*.

**Figure 2 sensors-24-07294-f002:**
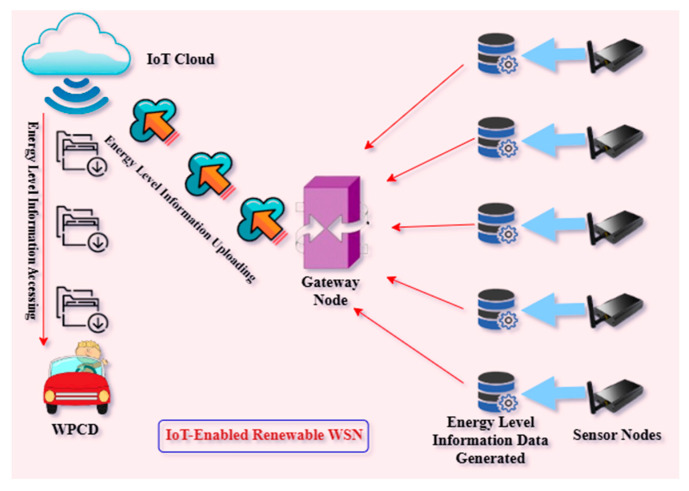
The remaining energy level is uploaded and accessed from the IoT cloud.

**Figure 3 sensors-24-07294-f003:**
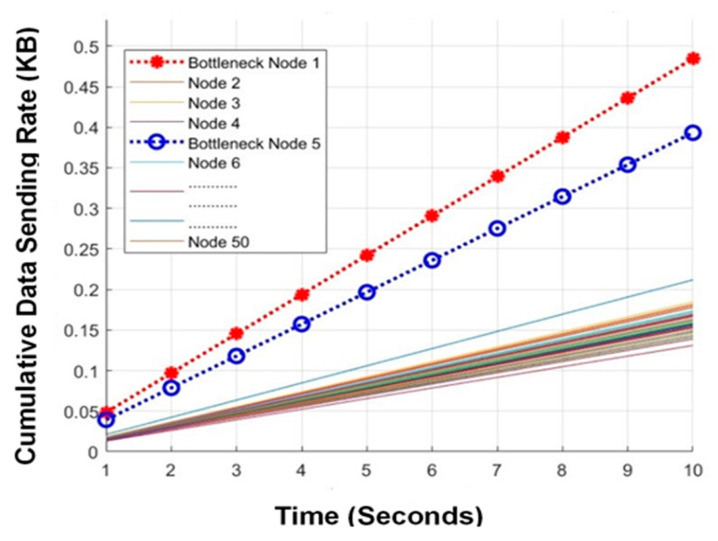
Data sending rate of bottleneck and other nodes.

**Figure 4 sensors-24-07294-f004:**
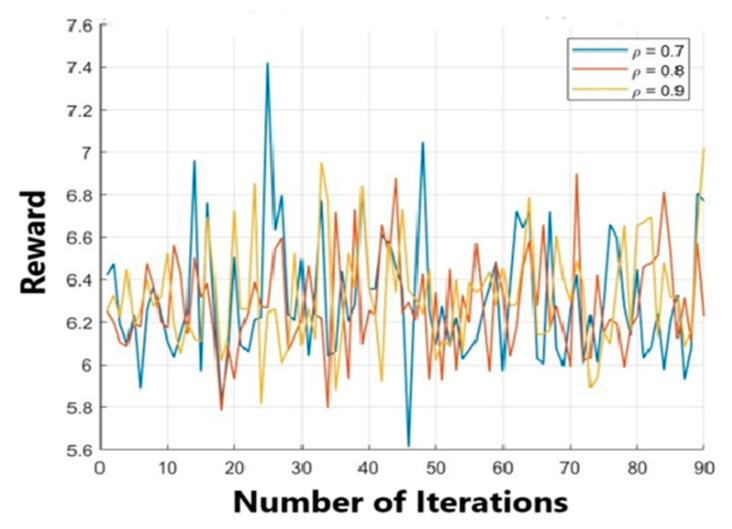
Reward function curve in WPCD-ACO.

**Figure 5 sensors-24-07294-f005:**
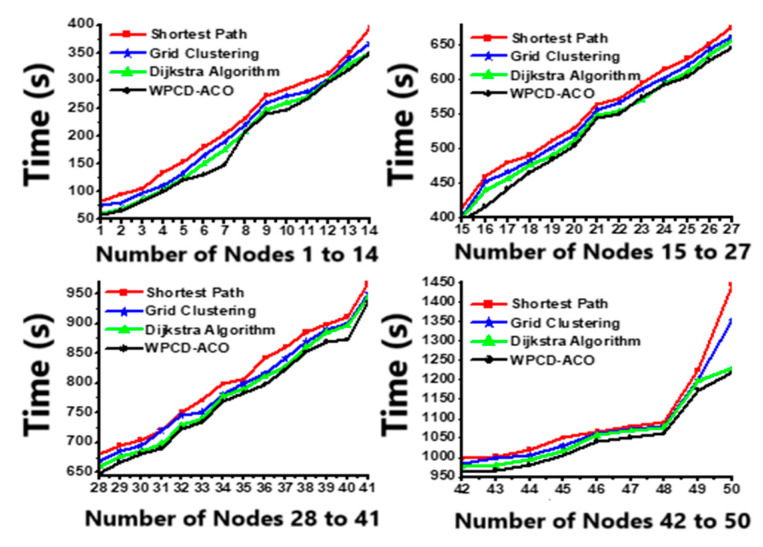
The arrival time of WPCD at each node is from 1 to 50.

**Figure 6 sensors-24-07294-f006:**
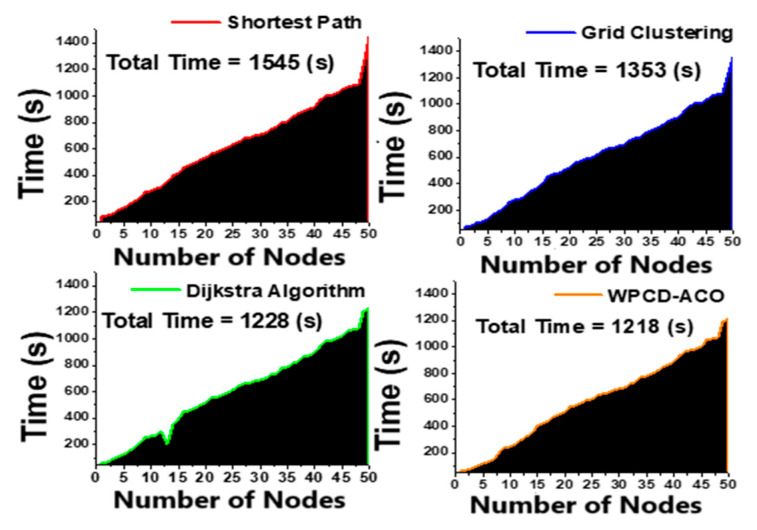
Total traveling time of the WPCD in the field.

**Figure 7 sensors-24-07294-f007:**
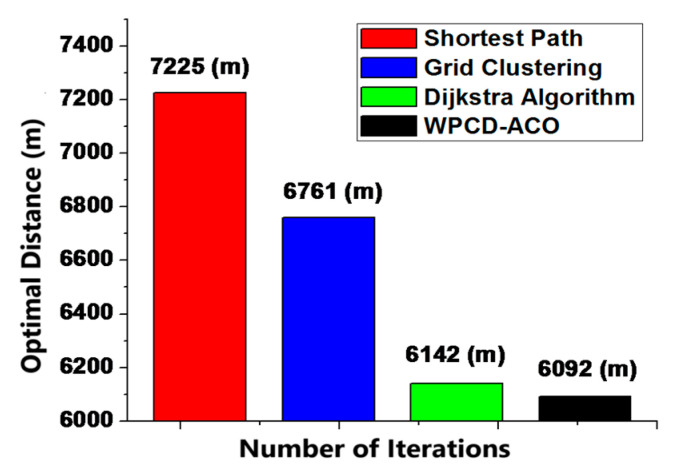
Optimal distance traveled by WPCD.

**Figure 8 sensors-24-07294-f008:**
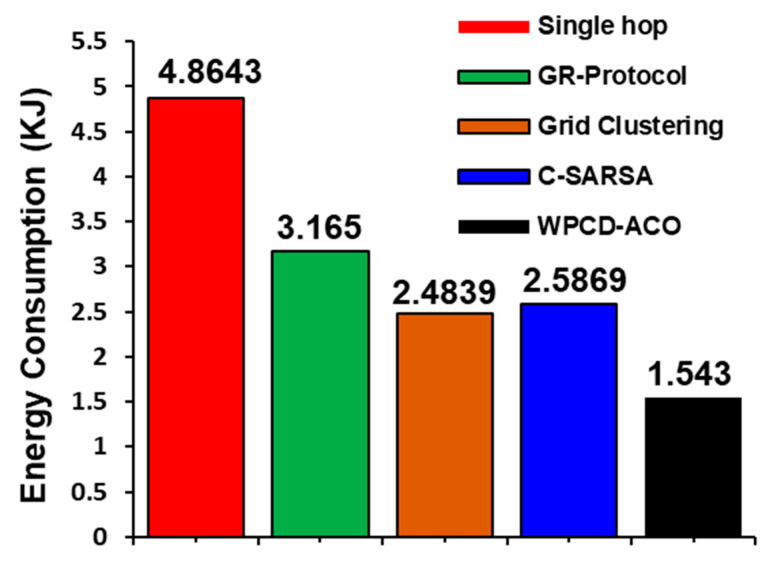
Total Energy consumption of RWSN.

**Figure 9 sensors-24-07294-f009:**
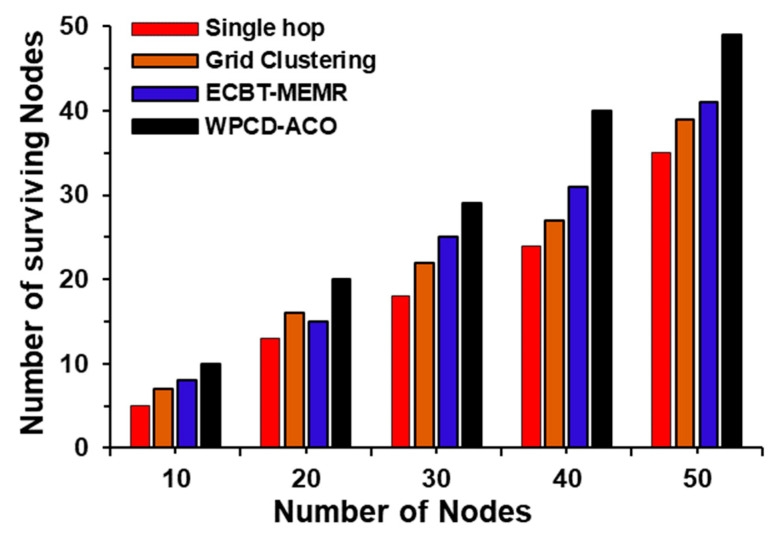
Number of surviving nodes in the RWSN.

**Table 1 sensors-24-07294-t001:** Notations of the system model.

Notations	Details of Notations
WPCD	charger (wireless portable charging device)
BS	base station
RS	rest station
Ρ	data receiving power consumption coefficient
U	power charging rate
V	traversing speed of WPCD
*Allowed*	set of sensor nodes that have higher remaining energy than $E_{min}$
*C_ij_*	energy consumption coefficient of sending data from node *i* to *j*
*C_iB_*	energy consumption coefficient of sending data from node *i* to BS
*prior*	set of nodes with lower remaining energy $E_{xmin}$
$E_{max}$	fully charge energy
$E_{min}$	minimum energy at which a sensor node can remain functional
${visited}_{k}$	the set of nodes which need to be visited by ants
*Bottleneck Node*	*Nb*
$E_{xmin}$	the remaining energy of the bottleneck node
$g_{ij}$	coefficient of data flow from node *i* to *j*
$g_{iB}$	coefficient of data flow from node *i* to BS
$R_{i}$	data generating rate
*τ*	complete time of the WPCD in the field
*τ* * _vac_ *	vacation time of the WPCD
$a_{i}$	approaching time of the WPCD of node *i*
*z* phase	specific period to recharge the lower remaining energy first
$D_{c}$	complete distance traveled by the WPCD

**Table 2 sensors-24-07294-t002:** Simulation parameters and their values.

Parameters	Values of Parameters
*N*	50
RS position	[0, 0]
BS position	[500, 500]
U	5 watts
Antenna	Omni-directional
Total ants A	20, 30
*h*	5 cm/s
V	5 m/s
Path loss	Log-normal shadowing
α	1
β	1
ρ	0.7
Q	1
Number of iterations	30, 60, 90
c	0.0014

## Data Availability

Data are contained within the article.

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
