# Peer review of "Intelligent Wireless Charging Path Optimization for Critical Nodes in Internet of Things-Integrated Renewable Sensor Networks"

_sensors, 2024, doi:10.3390/s24227294_

Round 1
Reviewer 1 Report
Comments and Suggestions for Authors
This paper addresses the challenge of bottleneck nodes by formulating an optimization problem that aims to identify the optimal traveling path for the WPCD, with a focus on minimizing energy consumption and enhancing network stability. Authors introduce an adaptive routing algorithm for the WPCD based on Ant Colony Optimization. Some issues should be addressed as follows.
1. In the abstract, the background, motivations, and numerical results should be simplified, while the novelty and main contributions should be emphasized.
2. In the introduction, there is no need to provide ref. [1]-[6] as the applications. Besides, the logic of the paragraphs should be improved, i.e., the first paragraph is too long, and it is hard to see the clear logic of introducing contributions of references since authors adopt 22 references in a single paragraph.
3. As the background of this paper is wireless sensor networks (WSNs) and the Internet of Things (IoT), recent high quality works on wireless powered IoT should be introduced, such as Distributed DDPG-based resource allocation for age of information minimization in mobile wireless-powered Internet of Things, IEEE IoTJ.
4. For the related work section, authors are suggested to update some outdated references by recent high quality works on the WSNs and IoT.
5. What is the significance of considering the threshold value? In my point of view, its value might be different for nodes.
6. It is obvious that the range “Emax<=i ” is incorrect.
7. Writing issues should be checked throughout this paper, i.e., i in the “The battery capacity of apiece node i ” should be written in the formula format. The format of (3) should be revised.
8. The details, writing, and format issues of the theoretical analysis should be improved.
9. Why authors adopt the ACO for the formulated problem? The relationship between the formulated problem and the ACO solution is not clear.
10. How does algorithm 1 advance in the research field? Does it obtain the global optimal path as step 15?
11. Punctuation should be added in formulas and steps of the algorithm.
Comments on the Quality of English LanguageThe English could be improved to more clearly express the research.
Author Response
We have carefully addressed all the comments and suggestions made by reviewer 1, which we believe have significantly enhanced the quality and clarity of our manuscript. We believe these revisions have strengthened our manuscript and made it more suitable for publication in the respective journal. We appreciate the reviewers' valuable insights, which have guided us in refining our work.
"Please see the attachment."
On behalf of all authors,
Dr. Nelofar Aslam

Reviewer 2 Report
Comments and Suggestions for Authors
The abstract is lengthy and would benefit from a more concise summary of the overall achievements. Details of the methodology should be omitted, as the abstract’s purpose is to provide a high-level overview rather than specifics on methodology.
In introduction section the classification of previous wireless charging methods is not entirely accurate. For example, categories such as drones serving as chargers and networks with multiple chargers should be placed under the same classification, as both involve the use of multiple charging sources, unless you are saying there will be a single drone in first method.
The list of contributions could be clarified, as sometimes the impact of a contribution is mistakenly listed as a contribution itself. A clearer structure will make it easier to understand the distinct contributions.
In the literature review, it is unnecessary to explain basic principles like magnetic coupling; instead, this section should focus on thoroughly reviewing existing methods with an emphasis on both their advantages and limitations. While various studies are discussed, the literature review lacks critical analysis to highlight the limitations of existing methods effectively. This should be extended and conducted more robustly so you do not miss any existing methods.
In Section 3 on Page 4, it would be valuable to explain why a two-dimensional space was chosen over a three-dimensional one, as real-world applications typically involve 3D spaces, especially considering a Wireless Power Charging Device (WPCD).
Additionally, in Section 3, Line 176 on Page 4, the sentence, “The battery capacity of apiece node i is defined under the…” should be rewritten to ensure clarity, particularly in distinguishing "i." Consistency in notation for node "I" is also needed, as it is sometimes italicized (e.g., Page 6, Line 207) and at other times not (e.g., Page 6, Line 209). Similarly, the parameter "z" should be formatted consistently, as seen on Lines 219 and 220.
For figure labels, the x-axis in Figure 4 should state "number of iterations" rather than "maximum iterations run." Likewise, the axes in Figures 5, 6, and 7 should be accurately labeled to improve clarity for the reader. Additionally, the sizes of the result figures should be consistent throughout the document for a more cohesive appearance.
The conclusion section could be enhanced by focusing on the major achievements of this work, ideally quantified, and aligning these with the stated list of contributions. Lastly, Reference 8 appears inconsistent with the formatting of other references. Please review all references to ensure they follow a uniform style across the document.
Author Response
We have carefully addressed all the comments and suggestions made by reviewer 2, which we believe have significantly enhanced the quality and clarity of our manuscript. We believe these revisions have strengthened our manuscript and made it more suitable for publication in the respective journal. We appreciate the reviewers' valuable insights, which have guided us in refining our work.
"Please see the attachment."
On behalf of all authors,
Dr. Nelofar Aslam

Round 2
Reviewer 1 Report
Comments and Suggestions for Authors
Authors have well addressed my previous comments.